# Revitalizing General Practice: The Critical Role of Medical Schools in Addressing the Primary Care Physician Shortage

**DOI:** 10.3390/healthcare11131820

**Published:** 2023-06-21

**Authors:** Christian J. Wiedermann

**Affiliations:** 1Institute of General Practice and Public Health, Claudiana—College of Health Professions, 39100 Bolzano, BZ, Italy; christian.wiedermann@am-mg.claudiana.bz.it; 2Department of Public Health, Medical Decision Making and Health Technology Assessment, University of Health Sciences, Medical Informatics and Technology, 6060 Hall in Tirol, Austria

**Keywords:** primary care, medical personnel shortage, medical education, curriculum modernization, new medical schools

## Abstract

The shortage of primary care physicians is a growing crisis that threatens the stability and effectiveness of healthcare systems. This paper explores a multi-pronged approach to addressing this issue by focusing on the modernization of medical curricula, the establishment of new medical schools, fostering collaboration between institutions, and implementing policy innovations. The cases of South Tyrol, Italy, and Tyrol, Austria, are examined to highlight the challenges faced in establishing new medical schools. This paper proposes that a comprehensive strategy, including the incorporation of general practice content and experience in medical education, is crucial for preparing future physicians for careers in primary care. Furthermore, intensifying efforts to establish new medical schools, particularly in regions such as South Tyrol, which lack native-language medical university education, can provide additional benefits in addressing regional needs and augmenting the number of graduates. Collaboration between existing and new medical schools, regional partnerships, and policy innovations are essential to support the establishment of institutions with a particular focus on general practice and the modernization of curricula at existing universities. By embracing this approach, stakeholders can collectively shape the medical education landscape and address the growing crisis of physician shortages in primary care.

## 1. Introduction

General practice is the cornerstone of healthcare, serving as the first point of contact for patients seeking medical attention [1]. However, a growing crisis threatens the stability and effectiveness of this vital component of the healthcare system, namely a significant shortage of primary care physicians. This shortage has serious implications for patient care and the overall health of communities that are summarized in Table 1 [2]. By providing a comprehensive overview of the crisis in general practice and the factors contributing to the shortage of medical personnel, this proposal aims to raise awareness of the urgency of this issue and stimulate discussion on potential solutions. Addressing this crisis is vital to ensure the continued availability of high-quality primary care for communities and to safeguard the overall health of the population.

## 2. The Role of Medical University Systems

In numerous nations, the primary issue revolves around the escalating scarcity of general practitioners (GPs), particularly in rural regions and other areas with inadequate services. While the total number of doctors per capita has risen in every country, the proportion of GPs has declined in the majority. As of 2020, a mere 20% of doctors were GPs, on average, throughout the European Union countries [3].

Medical universities have a significant influence on shaping the career choices and expertise of future physicians [4,5,6]. In 2022, a cross-sectional survey of medical students and resident doctors in Tyrol, Austria, to assess their attitudes and views on general practice revealed that both groups felt underprepared to work as a GP. The attractiveness of specialist medicine over general practice is related to additional factors, such as clearer content boundaries, better career opportunities, higher incomes, and research possibilities [4].

Medical-center-focused curricula in many medical universities may inadvertently contribute to a shortage of medical personnel in general practice [7]. By emphasizing specialized and hospital-based care, these curricula can overshadow the importance of general practice and limit students’ exposure to the reality of primary care [8]. Addressing this issue requires a reevaluation and modernization of medical curricula to support general-practitioner identity formation [9], better incorporate general practice, and encourage more students to pursue careers in primary care [10].

The Wass Report [11], titled “By choice not chance: Ensuring the future workforce of general practice”, examines the role of medical schools in promoting primary care as a career and addressing the shortage of primary care physicians. The report highlights the complex influences on medical students’ career choices and identifies significant issues impacting their perception of general practice, including professional tribalism, negativism within the profession, and financial challenges. The report emphasizes the need for collaboration among stakeholders and provides achievable recommendations across six stages of the student journey, aiming to enhance the understanding and representation of general practice as an academic specialty. It calls for collaborative leadership, resource optimization, and resolution of funding inequities to deliver a medical education experience that aligns with future healthcare needs and prepares students for careers in general practice [11].

By integrating general practice into medical curricula, students gain a comprehensive understanding of the healthcare system, including the critical role that primary care plays in promoting overall health and preventing diseases [6]. This holistic perspective helps students appreciate the value of general practice and its contribution to patient care and public health. It offers medical students the opportunity to work with a wide range of patients, from infants to the elderly, and manage various health conditions. This exposure helps students develop essential skills, such as effective communication, adaptability, and cultural competence, which are invaluable for any medical career. By providing students with positive experiences and role models in general practice, medical schools can foster an interest in the field, ultimately addressing the shortage of medical personnel in primary care [12]. A strong foundation in general practice is to equip medical students with the knowledge and skills needed to effectively collaborate with other healthcare professionals, such as nurses, pharmacists, and social workers. This interprofessional collaboration is essential for providing comprehensive and coordinated care to patients in increasingly complex healthcare environments.

### Modernizing Curricula at Established Medical Universities

To provide more primary care providers, medical schools must create an environment in which primary care is supported as a career choice. Medical schools should also consider educational models that incorporate regional campuses and rural educational settings [13]. The slow pace of change in curricula at many public medical universities can be attributed to a variety of factors, including complex bureaucratic processes and lengthy approval procedures that public universities often face, as these can hinder the implementation of systemic approaches in medical education [7]. Additionally, faculty members and administrators accustomed to traditional teaching methods may resist the adoption of new approaches, fearing a potential loss of expertise or status. Moreover, extrinsic factors, such as funding constraints and external regulations, can also influence the speed of curricular changes. Medical education has traditionally been organized in a hierarchical structure that prioritizes certain specialties and disciplines [14]. Changing the curricula to adopt a more systemic approach may challenge these established hierarchies, leading to resistance from faculty members who perceive a threat to their authority or expertise [10].

Implementing a systemic approach often requires significant investments in infrastructure, technology, and faculty development. Public universities, which are typically reliant on government funding, may struggle to secure the resources necessary to support these changes, especially in the face of competing priorities and budget constraints [7].

Although there is growing support for the adoption of systemic approaches in medical education, a consensus on the most effective methods and strategies is still emerging. This lack of agreement can make it difficult for universities to identify and implement the most appropriate changes to their curricula. While systemic approaches to medical education have been advocated for many years [1], robust evidence demonstrating their long-term impact on healthcare outcomes and workforce development is still limited. This lack of evidence makes it challenging for universities to justify significant investments in curricular reform.

## 3. The Challenges of Establishing New Medical Schools

In 2016, Switzerland increased the admission rate of medical students by 25 percent through new openings in five additional medical schools [15]. A new medical school could bypass bureaucratic hurdles and resistance to change that often plague established institutions, allowing for the development of innovative curricula that prioritize general practice and other systemic approaches [14]. Furthermore, a new medical school could attract faculty members who are committed to modernizing medical education, facilitating the establishment of a supportive and collaborative environment for innovation. However, starting a new medical school is not without challenges, including securing adequate funding, accreditation, and resources. Additionally, new medical schools may face opposition from the established institutions.

The commercialization of medical education has led to the emergence of new medical schools that are primarily focused on generating profits. These money-making institutions may prioritize their financial interests over the quality of education and training, which can negatively impact the development of competent and dedicated physicians [16]. While such schools may appear to offer quick solutions to the shortage of medical personnel in general practice, they may also compromise the quality of future healthcare providers, undermining the overall integrity of the medical profession. It is crucial that stakeholders in medical education and policymaking ensure that the establishment of new medical schools adheres to high standards of quality and ethics, prioritizing the improvement of healthcare outcomes over financial gain. In Austria, the Sigmund Freud Medical School faced the revocation of its accreditation due to the subpar quality of its master’s course [17]. This case sheds light on the financial conflicts of interest that arise in the education market for new medical schools, where the pursuit of profit and market competition can potentially compromise educational standards and the quality of programs.

### 3.1. Sociopolitical Factors and the Abandoned Medical School Initiative in South Tyrol, Italy

In South Tyrol, an autonomous province in Italy where the majority speaks German as a second national language [18] and health professionals need to be bilingual [19,20], the current shortage of health professionals is even more serious than in the rest of Italy [21]. This region faces unique challenges, including a lack of local medical schools, significant brain drain, and no access to German medical university education in Italy.

To address these issues, the South Tyrolean government proposed the establishment of a publicly funded medical school in 2012 [22,23]. However, to gain accreditation in Italy, this new institution needed to form a partnership with an existing Italian medical school [24]. This requirement became a contentious issue, as the rightist political opposition and some German-language media in South Tyrol strongly advocated against a trilingual medical school (Italian–German–English) and for maintaining ties with the German-speaking mother university of Innsbruck in Austria, where the majority of South Tyrolean medical students traditionally pursued their studies [25]. The opposition’s stance, rooted in cultural and linguistic concerns, effectively stalled the progress of the proposed medical school in South Tyrol [26]. The resistance to collaboration with an Italian partner, bolstered by the support of local media, ultimately led to the abandonment of the project.

This outcome underscores the complex interplay between sociopolitical factors and educational policies in shaping the landscape of medical education and its impact on addressing the shortage of medical personnel in general practice.

### 3.2. Institutional Resistance in the Medical School Initiative in Tyrol, Austria

In Tyrol, Austria, the local government recognized the falling number of students at its medical university and, thus, the impending personnel shortage in primary care [27] and commissioned a feasibility study for the establishment of a second medical school in addition to the existing state-run Medical University of Innsbruck in 2016 [28]. This initiative aimed to meet the region’s future healthcare workforce needs and enhance the overall quality of medical education.

The proposed project adopted a cross-border collaborative approach involving both Tyrol and South Tyrol, integrating most of the public hospitals and services within the Tyrolean and South Tyrolean Health systems [29,30]. The innovative curriculum focused on modern didactics, systemic teaching infrastructure, and admission criteria, with an emphasis on producing primary care physicians who would serve their local communities while still adhering to evidence-based practices. To enhance the proportion of graduates pursuing careers in primary care, medical schools should explore the implementation of a pre-matriculation program aimed at attracting and equipping motivated and skilled students with an interest in primary care. Admission committees should gain insights into the demographic factors linked to a higher probability of selecting primary care paths. Among these factors, an applicant’s expressed interest in primary care stands out as the most significant identifiable trait [31]. Independent institutions conducted thorough assessments, including legal consultations, workforce projections, architectural feasibility, and digital infrastructure for campuses and online learning and teaching. The resulting cost estimates were deemed financially viable. Throughout the planning process, the Medical University of Innsbruck was actively involved with various working groups.

However, despite comprehensive preparations, the project encountered opposition from Austria’s three public state medical universities at the time [32,33], including the Medical University of Innsbruck [34]. This resistance persisted, and ultimately the local government decided against proceeding with the establishment of a second province-run medical school focusing on modern primary care, after which the partner, South Tyrol, had also withdrawn from the original joint project [35].

In scientific terms, this case illustrates the challenges of implementing innovative medical education initiatives in the face of institutional resistance and complex sociopolitical dynamics. This underscores the need for strong collaboration and consensus building among stakeholders to overcome barriers and ensure the successful development of medical education programs that address the shortages of primary care physicians.

## 4. Potential Solutions

### 4.1. Modernizing Curricula in Existing State Medical Schools

To address the personnel shortage in general practice, existing state medical schools must undertake a comprehensive modernization of their curricula [13]. This entails the incorporation of general-practice content and experience throughout the educational program, with an emphasis on community-based and primary care settings. Integrating interprofessional education, preventive medicine, and early intervention strategies can foster a holistic understanding of healthcare among students. Furthermore, leveraging innovative teaching methodologies, such as problem-based learning, simulation, and digital technologies, can help engage students and enhance their clinical reasoning and decision-making skills [36]. By implementing these curricular changes, state medical schools can better prepare graduates for careers in general practice and contribute to a more robust primary care workforce.

### 4.2. Fostering Collaboration and Supporting New Medical Schools

Collaboration between established medical schools and new institutions focused on general practice can play a crucial role in addressing the shortage of medical personnel in primary care [37]. Established schools can share their expertise, resources, and best practices with new institutions, facilitating the development of high-quality educational programs that prioritize general practices.

In turn, new medical schools can bring new perspectives and innovative approaches to medical education, benefiting the broader academic community. Policymakers must recognize the potential of these collaborations and implement measures that support the establishment of new medicine curricula dedicated to general practice and rural medicine [38]. This could include streamlining the accreditation process, providing financial incentives, and encouraging partnerships with the existing schools. By fostering a collaborative and supportive environment, the medical education landscape can evolve to address the needs of primary care better and ensure the sustainability of the healthcare system.

The existing collaborations in South Tyrol, both with Cattolica University’s private medical school in Rome, Italy, for health professions [39] and with the private Paracelsus Medical University in Salzburg, Austria, for scientific collaboration [40], can serve as a foundation for the development of a new modern medical school in the region. The South Tyrol health service can leverage its existing relationships with Cattolica University and Paracelsus Medical University to develop a comprehensive plan for the establishment of a medical school in the region [41]. Drawing on the expertise of partner institutions, the new medical school can create a cutting-edge curriculum that prioritizes general practice and primary care quicker than established medical schools.

Partner institutions can collaborate to provide the necessary infrastructure and resources. They can also support the recruitment and training of high-quality faculty members, ensuring that the new school has access to a diverse range of medical-education expertise. The joint committee should work closely with relevant accreditation bodies and regulatory agencies to ensure that the new medical school meets all necessary requirements for accreditation and recognition, both nationally and internationally. Thus, by building on existing collaborations and utilizing the expertise of partner institutions, South Tyrol can establish a new medical school that addresses the region’s healthcare needs and contributes to a more robust primary care workforce.

### 4.3. Policy Innovations for Empowering New Medical Schools

Policymakers can consider implementing policy changes to support the establishment of new medical schools focused on general practice [42]. These may include simplifying and expediting the accreditation process for new medical schools that prioritize general-practice education, as well as financial incentives, such as grants, low-interest loans, and tax breaks, to encourage the development of new medical schools that focus on general practice. These incentives could help offset the initial costs associated with establishing a new institution and make it more attractive to potential investors or partners.

Collaboration between new medical schools and established institutions, such as universities or healthcare organizations, could involve creating formal partnership agreements, facilitating knowledge-sharing opportunities, or providing funding for joint initiatives focused on general-practice education [43].

Policies that support the integration of interprofessional education and collaboration within new medical schools could include funding for interdisciplinary training programs, encouraging the development of shared curricula, or mandating interprofessional education as a requirement for accreditation [44]. Resources and guidance for new medical schools to establish strong connections with local healthcare providers and community organizations could involve facilitating partnerships, offering financial support for community-based education initiatives, and creating policies that prioritize community engagement in medical education. A centralized body or database for collecting, analyzing, and disseminating best practices in general-practice medical education could provide new medical schools with valuable resources and guidance, promoting the continuous improvement and innovation of general-practice education across the sector [45].

By implementing these policy changes, governments can create a supportive environment for the establishment of new medical schools that prioritize general practice, ultimately contributing to a stronger primary care workforce and improved healthcare outcomes. Potential solutions are summarized in Table 2.

The solutions presented in this table provide general approaches to address the shortage of general practitioners. Region-specific solutions for South Tyrol can be found in Table 3. These solutions aim to address the shortage of general practitioners in South Tyrol and enhance primary care education.

## 5. Learning from the Past and Moving Forward

The experiences of South Tyrol, Italy, and Tyrol, Austria, highlight the challenges faced in establishing new medical schools and modernizing curricula to prioritize general practice. These cases emphasize the importance of engaging stakeholders from various domains and fostering open dialogue and consensus-building to facilitate meaningful change in medical education. By involving a diverse range of voices and perspectives, we can address the sociopolitical factors, bureaucratic hurdles, and institutional resistance that often hinder the implementation of innovative medical-education initiatives.

### 5.1. Collaborative Dialogues: Bridging Disciplines and Perspectives

One crucial aspect of fostering collaborative change in medical education is the active involvement of stakeholders beyond traditional healthcare professionals. This includes education specialists, literature professors, social workers, philosophers, clergy, and local non-profit organizations. By bringing together professionals from these diverse fields, we can create a platform for interdisciplinary dialogues that enrich the educational experience and promote a comprehensive understanding of primary care.

Education specialists contribute their expertise in pedagogical advancements and innovative teaching methodologies. Their input helps shape curricula that effectively prepare future physicians for the challenges of primary care, integrating practical skills, critical thinking, and lifelong learning strategies.

Literature professors play a vital role in emphasizing the importance of effective communication skills, empathy, and the ability to understand patients’ narratives and experiences. By exploring literary works that highlight the humanistic dimensions of medicine, students gain insights into the complexities of patient care and develop a deeper appreciation for the patient’s perspective.

Social workers offer valuable insights into the social determinants of health and the impact of socioeconomic factors on healthcare access. Their collaboration with medical schools allows for the integration of community-oriented approaches, fostering an understanding of the broader context in which primary care operates.

Philosophers provide a philosophical foundation for medical practice, encouraging students to reflect on ethical considerations, professionalism, and the values that underpin their future roles as physicians. Their perspectives promote critical thinking, ethical decision-making, and the exploration of the moral dimensions of primary care.

Clergy members contribute insights into the spiritual and psychosocial aspects of patient care, recognizing the significance of holistic approaches in promoting overall well-being. Their involvement helps foster a deeper understanding of the psychosocial factors that impact patients’ health and the importance of compassion and empathy in delivering patient-centered care.

Local non-profit organizations play a crucial role in bridging the gap between medical education and community needs. By partnering with these organizations, medical schools gain practical insights into community-based healthcare initiatives, allowing students to engage directly with the local population and address the specific healthcare challenges of the region.

Through collaborative dialogues that involve stakeholders from various disciplines, we can create an educational environment that fosters interdisciplinary learning, empathy, and a comprehensive understanding of primary care. By recognizing the valuable contributions of professionals from diverse fields, we enhance the relevance and quality of medical education, preparing future physicians to meet the complex healthcare needs of their communities.

In conclusion, engaging stakeholders from education, literature, social work, philosophy, clergy, and non-profit organizations is essential for driving collaborative change in medical education. Their expertise and perspectives enrich the educational experience, ensuring the holistic preparation of future physicians for primary care practice. By embracing interdisciplinary dialogues and fostering open dialogue and consensus-building, we can overcome barriers and create a medical education system that aligns with the evolving healthcare landscape.

### 5.2. Call to Action

Although the original concepts from Canada and Australia with a much stronger adaptation of curricula to primary care content have been proposed for more than 20 years to counter GP distribution disparity, including an increased admission of students from rural areas for later work in rural areas [49], the problematic situation today is essentially unchanged. As outlined here, to address the shortage of primary care physicians, it is crucial to modernize curricula at established medical universities. These should incorporate general-practice content and experience, emphasize community-based and primary care settings, and integrate innovative teaching methodologies. Collaboration between established and new medical schools supports the accreditation of new medical schools and curricula by sharing expertise, resources, and best practices to develop high-quality educational programs that prioritize general practice. Policy changes are necessary to streamline accreditation, provide financial incentives, and encourage partnerships with schools.

The engagement of stakeholders in open dialogue and consensus building helps address the complex challenges faced in establishing new medical schools and modernizing existing curricula. By embracing these calls for action, the medical education landscape can evolve to better address the needs of primary care and ensure the sustainability of the healthcare system.

## 6. Conclusions

The growing crisis of medical personnel shortages in primary care poses a significant threat to the stability and effectiveness of healthcare systems. Addressing this issue requires a multi-pronged approach that encompasses various aspects, including those of medical-education-modernizing curricula at established medical universities, intensifying efforts to establish new medical schools (especially in regions that lack local medical education institutions, such as South Tyrol), fostering collaboration between institutions, and implementing policy innovations to support these initiatives. This comprehensive strategy can effectively address the diverse challenges faced by the healthcare system and promote the development of a well-prepared primary care workforce. Modernizing existing curricula at established medical universities by incorporating general-practice content and experiences, emphasizing community-based settings, and integrating innovative teaching methodologies can better prepare future physicians for careers in primary care, ultimately helping address the shortage of medical personnel in general practice.

However, the establishment of new medical schools offers additional advantages beyond curriculum modernization and improved didactics. New medical schools can accommodate more students, thus increasing the number of graduates, which is something that state universities may be limited in due to financial constraints. Additionally, access to new schools can be regulated to address regional needs by considering factors such as social characteristics. South Tyrol, in particular, faces challenges due to its dependence on foreign-state medical schools in the German language, as there are no local medical schools in the region. In contrast to Tyrol, which already has the Medical University of Innsbruck, efforts to establish a new medical school in Southern Tyrol should be intensified. This would help to address the unique challenges faced by the region and create opportunities for curriculum innovation and regional access to medical education.

In summary, while modernizing existing curricula remains crucial, intensifying efforts to establish new medical schools, particularly in regions such as South Tyrol, can provide significant additional benefits in addressing the needs of primary care and ensuring the sustainability of the healthcare system. By embracing this call for action, stakeholders can collectively shape the medical education landscape and better address the growing crisis of medical-personnel shortages in primary care.

## Figures and Tables

**Table 1 healthcare-11-01820-t001:** Tackling the shortage of medical personnel in primary care.

Topic	Description
Crisis in General Practice	
Importance of General Practice	Primary care and preventive medicine
	Early detection and management of health conditions
	Continuity of care and building long-term patient relationships
Shortage of Medical Personnel	Declining interest in general practice as a career choice among medical students
	Increasing patient populations and demand for primary care services
	High workload and burnout among general practitioners
	Impact of the shortage on patient care, waiting times, and healthcare costs
Factors Contributing to the Shortage	
Medical Education and Training	Medical university system’s focus on specialized and hospital-based care
	Insufficient exposure to general practice during medical education
	Limited availability of postgraduate training positions in general practice
Socioeconomic Factors	Perceived lower prestige and financial rewards compared to specialist roles
	Geographic disparities in the distribution of medical personnel
	Impact of an aging population and increasing prevalence of chronic diseases
Policy and System-Level Factors	Insufficient support for general practice from policymakers and healthcare organizations
	Barriers to establishing new curricula and medical schools focused on general practice
	Resistance from established medical universities

**Table 2 healthcare-11-01820-t002:** Potential solutions for addressing general-practice shortage.

Solution	Description
Modernizing Curricula [46]	Incorporate general-practice content and experiences throughout the educational program
	Emphasize community-based and primary care settings
	Integrate interprofessional education, preventive medicine, and early intervention strategies
	Employ innovative teaching methodologies, such as problem-based learning, simulation, and digital technologies
Fostering Collaboration [47]	Encourage collaboration between established medical schools and new institutions focused on general practice
	Share expertise, resources, and best practices
	Implement measures that support the establishment of new medical schools dedicated to general practice, such as streamlining the accreditation process, providing financial incentives, and encouraging partnerships with existing schools
Policy Innovations [48]	Simplify and expedite the accreditation process for new medical schools prioritizing general practice education
	Offer financial incentives, such as grants, low-interest loans, or tax breaks
	Foster partnerships between new and established institutions
	Support integration of interprofessional education and collaboration
	Provide resources and guidance for community-based education and engagement
	Establish a centralized body or database to collect, analyze, and disseminate best practices in general practice medical education

**Table 3 healthcare-11-01820-t003:** Region-specific solutions for addressing general-practice shortage in South Tyrol.

Solution	Description
Developing Regional Partnership	Leverage existing collaborations between South Tyrol health service, Cattolica University, and Paracelsus Medical University
	Create a comprehensive plan for establishing a medical school in the region
	Develop a cutting-edge curriculum prioritizing general practice and primary care
	Collaborate on infrastructure, resources, and faculty recruitment
	Ensure accreditation and recognition at national and international levels

## Data Availability

No new data were created.

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
