# Peer review of "Revitalizing General Practice: The Critical Role of Medical Schools in Addressing the Primary Care Physician Shortage"

_healthcare, 2023, doi:10.3390/healthcare11131820_

Round 1
Reviewer 1 Report
The essay addresses the important issue of the shortage of primary care physicians and proposes several helpful solutions which will be interesting to healthcare administrators and educators.
The only thing that needs improvement is the tables, which are very difficult to read. To improve, please add a few more divisions or lines so that the connections between topics and descriptions are clear.
I re-emphasize the need for the tables to be formatted in a clear line by line manner. As they currently are, the tables are very difficult to understand. Maybe reformatting involves just adding horizontal lines. Maybe it also means aligning the text to the left. Changes to the tables are necessary.
An area that could be strengthened is Section 5: Learning from the Past and Moving Forward. Each of the paragraphs here could be expanded to add illuminating details and explanations even if much of what is added has already been discussed in earlier sections.
For example, what does the author mean by "open dialogue and consensus-building"? How is this shown in Tyrol? Who are some of the stakeholders the author has in mind? The author discusses open dialogue between medical institutions and other healthcare professionals, but what about others, such education specialists, literature professors, social workers, philosophers, clergy, and local non-profit organizations?
Reviewer 2 Report
I enjoyed reading this article. The shortage of primary care physicians is an important international issue so is likely to be of interest to a wide readership. This article is well-written and strikes a good balance between the general issues and the specific problems encountered when trying to address this in Austria and Italy. In the UK the 'Wass Report' (By choice not chance) performed a detailed review of the influence of medical schools on subsequent career choice of general practice, and I was a little surprised not to see it referenced here instead of some older references.
I found the layout of the tables unhelpful, which made them hard to read. It was difficult to work out which description related to which topic or solution. I would be grateful if this was addressed. It might also be useful to add supporting references to the proposed solutions to help an interested reader easily find further reading.
Reviewer 3 Report
Abstract: clearly described problem (shortage of GPs); aim of paper also clear: highlighting what can be done to ameliorate the situation regarding lack of GPs with particular reference to South Tyrol.
Very neat the way ChatGPT has been used: good innovation I will consider for the future!
Paper
L56: ‘medical identity formation’ is too general; general practitioner identity formation might be better?
L80: ‘slow pace of change in medical curricula’: you imply that it is all the fault of the universities. What about extrinsic factors?
L138: my ignorance (and probably other readers if not German or Italian): why trilingual? Not bilingual?
I would prefer the tables to be in boxes as they are slightly difficult to read as they are.
For table 2 I would just put in general potential solutions and then have a table 3: solutions specific to the South Tyrol.
Could references not in English be translated? As the rest of the paper is in English.
Overall I found this to be a very well-written article, hightly relevant to a topic which has become a problem in many countries. I think the suggestions for potential solutions are sound and will interest many educators in countries/areas where the provision of GPs is problematic.
The English in the paper is fine, but I would like to see all the references in English, rather than a mix including German and Italian.
